# Contrasting the Practices of Virus Isolation and Characterization between the Early Period in History and Modern Times: The Case of Japanese Encephalitis Virus

**DOI:** 10.3390/v14122640

**Published:** 2022-11-26

**Authors:** Goro Kuno

**Affiliations:** Centers for Disease Control and Prevention, Formerly Division of Vector-Borne Infectious Diseases, Fort Collins, CO 80521, USA; gykuno@gmail.com

**Keywords:** Japanese encephalitis, Japanese encephalitis virus, Nakayama strain, concept of new virus isolation, transmission mechanism, metagenomics, phylogeny, host range determinant, voucher system, virus species criteria

## Abstract

Japanese encephalitis is a serious disease transmitted by mosquitoes. With its recent spread beyond the traditional territory of endemicity in Asia, the magnitude of global threat has increased sharply. While much of the current research are largely focused on changing epidemiology, molecular genetics of virus, and vaccination, little attention has been paid to the early history of virus isolation and phenotypic characterization of this virus. In this review, using this piece of history as an example, I review the transition of the concept and practice of virus isolation and characterization from the early period of history to modern times. The spectacular development of molecular techniques in modern times has brought many changes in practices as well as enormous amount of new knowledge. However, many aspects of virus characterization, in particular, transmission mechanism and host relationship, remain unsolved. As molecular techniques are not perfect in all respects, beneficial accommodation of molecular and biologic data is critically important in many branches of research. Accordingly, I emphasize exercising caution in applying only these modern techniques, point out unrecognized communication problems, and stress that JE research history is a rich source of interesting works still valuable even today and waiting to be discovered.

## 1. Introduction to the Need of Re-Examining the History of Japanese Encephalitis

Japanese encephalitis virus (JEV) is the most serious cause of encephalitis in humans among mosquito-borne viruses [1,2]. Currently, nearly 3 billion people in Asia are at risk of Japanese encephalitis (JE), with the estimated symptomatic cases ranging between 55,000 and 75,000 per annum. Because of the seriousness in public health, it has been dubbed “the plague of the Orient” [3]; but it has begun to shed its regional mantle, as it caused occasional outbreaks in the Pacific islands and spread to Australia and even to Pakistan [4]. Apparently, the virus may have spread to Italy and Angola, since the viral RNA fragments have been detected there [5,6]. Still, more field work is necessary to confirm these discoveries in Africa and Europe in the absence of virus isolation since detection. On the other hand, its incursion deep into southern parts of Australia in 2022 has alarmed not only public health sector but veterinary industries there [7].

The above examples of geographic spread of JEV, including repeated spread to Japan presumably from continental parts of Asia [8,9], may be explained by a new discovery of the involvement of a subtype of the major vector, *Culex tritaeniorhynchus*, endowed with a capacity of long distance dispersal [10]. Thus, it would be interesting to learn if this subtype of vector was also involved in the latest spread of JE to southern parts of Australia in 2022. 

Actually, this long-distance spread of JEV may not be a recent event, according to one hypothesis. Puzzled by the fact that St. Louis encephalitis virus (SLEV) is the sole member of the JEV Complex among mosquito-borne flaviviruses in the New World, Albert B. Sabin speculated a possibility that JEV’s progenitor in the Old World could have spread to the New World to evolve into SLEV [11].

As evidence of long distance dispersal from continental parts of Asia to Japan mounted, re-evaluation of the early history of JE in Asia has become necessary for multiple reasons. First, if the first recognition of JE in Japan in 1871 was accurate, most likely something happened to the populations of the vector mosquitoes in relation to their dispersal from tropical Asia (the presumed birth place of the virus) and/or to the ecologic conditions in Japan (such as increase in swine breeding). While re-examination of the early JE research history in Japan is one of the major objectives of this review, re-examination of the Chinese history is equally important for the following reasons.

According to a molecular clock determination in one study, JEV is estimated to have evolved around 3255 years ago [12]. Though a doubt about the accuracy of this clock remains, assuming that it was correct, a serious question emerges about the possibility of much earlier records of JE or JE-like diseases in China because the conditions in ancient China were actually more ideal than in Japan. First, horses were indispensable in life and abundant; thus, they were revered for their enormous value in Chinese history. Consumption of pig meat has been a long tradition. The geographic proximity of southern parts of China to the presumed birthplace of JEV could have facilitated repeated dispersal of infected vectors through constant human movement, trade and cultural interactions over centuries. Thus, the conditions were ideal for JE activity. On top of these favorable conditions, the richness of archival medical records in China is well known. Accordingly, if one examines carefully the infectious disease history there, there is a possibility of finding outbreak records of equine (or human) encephalitis and stillbirth in pigs due to JEV or similar neurotropic agent much earlier than 1871 when JE was recorded in Japan for the first time. Furthermore, the evidence of repeated introduction of JEV to Japan from somewhere in continental parts of Asia and confirmation of air-borne eastward movement of mosquitoes across the East China Sea made nearly 4 decades ago further support a need for the exploration of archival medical records in China.

On the other hand, if total absence of any hint of JE outbreak in the long history of China was correct, then it would more strongly suggest that vector subpopulations endowed with long-distance dispersal or more vector-competent subtypes only emerged around 1871. The studies along this line of thought may also explain why JE has not occurred in many parts of Middle East and Africa where the major vector of JEV (*Culex tritaeniorhynchus*) probably has existed for centuries. Accordingly, any explanation for the recent detections of JEV RNA in Angola and Italy would become an intertwined subject of interest regarding the mechanism of JE spread. 

The following medical puzzle also deserves further historical research. Outbreaks of JE were early recorded in East Asia (Japan, Korea, China, and Far East part of Russia) but recorded in tropical Asia much later. However, phylogenetic studies established a chronologic opposite: that JEV evolved in tropical parts of Asia first and subsequently spread elsewhere including East Asia. This raises another set of questions. Why was not outbreak or JE-like neurologic case recognized in the area of virus origin for so long? Was the virulence of the early virus population benign but a virulent subpopulation emerged much later? Or was early virus population almost exclusively confined to wildlife but zoonotic pattern of transmission emerged only much later? A recent revelation of JE outbreaks in humans and horses in Malaysia as early as 1942 [13] may necessitate revision of the timeline of JE history in tropical parts of Asia. 

In contrast to other regions of Asia where early records of JE-like outbreak are scarce at present, the amount of documents that needs re-examination in Japan is very large, as revealed in this review. However, the subject is very broad and the number of documents enormously large. Accordingly, I provide a glimpse into the history by focusing only on a few facets of the historical subjects, “first” isolation of JEV, virus persistence and early characterization of viral transmission. The reason for focusing on the first facet is as follows: when the JE literature awash with exciting new discoveries and other developments is examined, three dates (1924, 1933, and 1935) of “first” JEV isolation are found, indicating a state of confusion among authors. Assuming that 1924 was correct, finding out the exact cause of this confusion would be timely, as the centennial of the “first” JEV isolation is approaching fast. The reason for the selection of the second and third facets is to provide readers many early contributions that have been rarely revealed to the world despite their importance in current JE research, largely due to language barrier and other circumstances.

First, I review the history of the “first” JEV isolation in Japan but capture the topic in the contexts of the basic concepts related to the discovery of new viruses and the requirements that must be met for the discovery reporting. In a later section, I comment on the contrasting application of metagenomic technique for virus isolation. Using a similar comparative style between the early period and modern period, the status of virus characterization in two periods is compared, in particular with respect to our understanding over virus persistence, transmission mechanism, and host relationship.

Regarding the usage of terms, I use “strain” almost synonymously with “isolate,” although differential usage of the two terms is now recommended [14]. For the descendants from one strain, I use “strain derivative.” This is because accepting new definition of strain or other nomenclatures below the species level adds more confusion to the interpretation of old documents. Additionally, “first” (within quotation marks) virus isolation refers only to the isolation of a new species of virus for the first time in history, while first (without quotation marks) virus isolation merely refers to isolation of virus from a particular host, vector, vertebrate, place or anything else for the first time. In addition, “host” is used to refer to vertebrates. As for the viruses covered, flaviviruses including JEV are the foci of this article, though other groups of viruses are mentioned infrequently to enrich discussion. 

## 2. Method

Because most early JE references documented in Japanese, German and French by Japanese are not compiled in any of multiple online databases, relevant articles were searched manually and retrogressively through examination of the references cited, beginning with reviews published before 1960. The identified articles were retrieved from the National Diet Library, Tokyo. Many references written in English and others in German were found online through Google search. Other originals were obtained from the Library of University of Notre Dame, Notre Dame, Indiana.

## 3. Concept and Requirements for “First” Virus Isolation

### 3.1. Concept

New species of viruses are constantly discovered. Some of them are isolated in mission-oriented research, in particular in etiologic investigations of known diseases. The history of JEV isolation falls in this category. Thus, historically, virus discovery has been biased towards the agents involved in known diseases. With the modern method of virus discovery by means of a combination of metagenomics and bioinformatics, while objectives of some are mission-oriented, for others the objectives are a mixture of exploration and/or survey. It is speculated that more than 99.9% of the virosphere remains undiscovered. 

Viruses have been also discovered accidentally during research for other objectives. Some viruses of unknown significance were dubbed “viruses in search of a disease” or “orphan virus.” One of such viruses was Cell-fusing agent virus [CFAV] (a flavivirus) discovered accidentally in 1975 in the Joseph Peleg’s *Aedes aegypti* cell culture [15]. Because Peleg used the Rockefeller Institute’s colony of *Ae. aegypti* (“Rock” strain for Rockefeller) to establish his cell line and because the colony there had derived from Carlos Finlay’s original colony presumably established circa 1880 in Cuba [16], occurrence of CFAV in *Ae. aegypti* populations in the Caribbean was suspected and was indeed confirmed later by its isolation from this mosquito in Puerto Rico [17]. 

With the advent of metagenomic techniques, more commensalistic viruses were discovered, and some new viruses were found to be even beneficial to the hosts. Thus, establishing a causal relationship as a requirement for the newly discovered viruses partly became unimportant or irrelevant; and further revision of already-modified Koch’s postulates became again necessary in the eyes of modern virologists, since many new viruses cannot be propagated [18]. 

The scientists who discovered viruses or isolated viruses of notable significance have received a special accolade, and their contributions have been celebrated in virology and/or in medical history books [19,20,21]. With the coveted honor at stake, special recognition or respect bestowed has been a source of pride for the recipients of gratitude and attention. 

Accordingly, dispute over who isolated a particular virus for the first time in history has erupted occasionally, such as in the discovery of human immunodeficiency virus in a form of highly open public argument or in the discovery of Ebola virus in a form of more subtle discussion. As shown in the following paragraphs, the “first” isolation of JEV superficially resembles the famous retrospective debate over who was first to discover virus as a new disease entity distinct from microorganisms, Ivanovskii, Beijerinck, Loeffler and Frosch or someone else? [22]. One common cause of the eruption of debate was different understanding among scientists about the requirements that must be satisfied to prove “first” virus isolation. However, one major contrast of the early JEV isolation history to this famous retrospective debate in Europe was that among early JEV workers such a debate over credit during the competitive period between 1920 and 1940 has never been known to have occurred, at least as far as public knowledge was concerned. Another contrast was that the entire history, except for a small number of reports published in English, has never been fully revealed to the world and regrettably forgotten. 

### 3.2. Requirements for “First” Virus Isolation

The understanding of the requirements for “first” virus isolation evolved by trials and errors over years, since little was known in the early part of history about the characteristics of viruses. Further, techniques used at first were rudimentary when even syringes had not been invented. The following requirements were a retrospective compilation, which means that at any given stage of JEV isolation period researchers recognized these criteria only partially. 

It should be noted that in historical accounts, sometimes “virus isolation” has been used for the events of “first recognition of viral etiology” or “first virus detection”; while “virus discovery” has been loosely used for virus isolation, virus detection and/or recognition of viral etiology. These ambiguous usages of words also contributed to controversies. With respect to the history of virus isolation and techniques used, an ebook available free online [20] is very useful. 

#### 3.2.1. Filterability and Submicroscopic Size

Development of unglazed porcelain filter by Charles E. Chamberland in 1884 for proof of viral etiology was unquestionably a major landmark. It was in 1924 when the etiology of JE was confirmed by this method. However, the conclusion was not without a controversy because some claimed evidence of bacterial etiology. This problem was not uncommon at the time because it was a time of transition from the age of bacterial to newly emerging viral etiology of infectious diseases. 

In modern times, the requirement of submicroscopic size was weakened by the discoveries of giant viruses (such as mimiviruses) some of which are microscopically visible; and requirement of filterability was violated by the discoveries of non-viral, filterable, and infectious entities (such as mycoplasmas, viroids, and prions). 

#### 3.2.2. Absence of Replication in Culture Media

This was found early as a good indication of viral agent, but debate persisted for some time because a small number of microorganisms that grow in media are filterable. Nonetheless, for JEV isolation, this was an important criterion. The absence of replication in culture media may suggest automatically that early virologists recognized obligatory need of living cells for viral replication. However, that interpretation was not necessarily uniform among early workers. 

#### 3.2.3. Identification of Optimal Source (s) in the Body of Infected Animal (or human) for Sampling, and Timing and/or Frequency of Sampling

Most early workers heavily relied on trials and errors and careful observation of diseased animal or human for this information. For JE, initially sampling sources for virus isolation were broad and included most major organs, the brain, cerebrospinal fluid, blood, saliva, nasopharyngeal secretion, urine, and feces. Though it was claimed by some that JEV was isolated from blood, it was not considered an optimal source, because of too many negative results. Ultimately, central nervous system, in particular, the brain, was chosen for isolation by most workers. For other sources of the virus, direct transmission was considered, as described in Section 5.3.

#### 3.2.4. Selection of Optimal Animal or Organ/Tissue Culture for Virus Propagation and Passage

In the 1920s and early 1930s, Asian monkeys, guinea pigs, and rabbits were used by many. Some used even dogs. The low rate of success in producing a neurologic symptom consistently over passages was a major problem, as the rate of claimed success varied considerably among workers. 

Cell/tissue culture technique was developed extensively since the early research in the 1910s by Alexis Carrel and many others. By the time of nationwide JE outbreak of 1935, use of chick embryo developed by Ernest W. Goodpasture in 1931 was added to the list of choices available for JEV isolation [23]. More importantly, German (or Swiss) albino mice that had been found very useful for the replication of St. Louis encephalitis virus (SLEV) a few years earlier by Leslie T. Webster at the Rockefeller Institute of Medical Research [24] were available. The result was spectacular. The use of these mice solved all past problems and resulted in a large number of isolations. Chick embryo organ or tissue culture technique was, however, more often applied not for virus isolation but for the propagation after isolation [25,26].

#### 3.2.5. Proof of Cause-Effect Relationship

In the 1920s and 1930s, when little was known about the factors determining the development of a sign, symptom or syndrome of JE, variations in animal species chosen, its age, dose, route of inoculation, and other factors enormously complicated the establishment of cause-effect relationship and delayed discovery of JEV. Appearance of a symptom or syndrome in experimental animals quite different from that in humans was other complication. 

Nevertheless, most investigators recognized the importance of establishing a cause-effect relationship with the agent isolated. Initially, reproducibility of the development of a neurologic symptom in experimental animals was thought to be sufficient to fulfill this requirement for JEV. However, the results were variable. To improve reliability, workers emphasized in vivo neutralization test (NT) using mice and homologous antiserum-virus mixture [27] because it generated more consistent results [28]. The factors that distorted the outcome of NT were asymptomatic infection, age of mice, and use of outbred mice. 

Causal relationship was also determined by immunologic (later serologic) tests. Among arboviruses, it was determined first for yellow fever virus with complement fixation (CF) test [29], followed by in vivo neutralization (NT) test [30]. CF test was then used for other arboviruses [31] and was later adopted for JEV [32,33]. On the other hand, hemagglutination-inhibition (HI) test, which had been developed by George K. Hirst for influenza diagnosis [34], was not used for JE diagnosis before 1950.

#### 3.2.6. Comparison with Known Viruses with Shared Characteristics

Though this is one of the most important requirements today, in the early part of history (before 1930), it was rarely performed simply because of the paucity of similar viruses isolated. 

As virus isolation activities accelerated in early 1930s, by the time JEV was isolated in 1935 this was a requirement for the confirmation of new virus isolation. However, the kinds of viruses to compare with varied depending on the inventory of viruses available in each laboratory. Thus, in Japan St. Louis encephalitis virus (SLEV) and Rift Valley fever virus were used for JEV, while in the U.S., SLEV was compared not only with JEV but with western equine encephalitis virus, Russian spring-summer encephalitis virus, and other neurotropic viruses [31,35]. 

#### 3.2.7. Preservation of Infectivity of Virus Isolate and Reproducibility 

Preservation of infectivity of isolated virus was a major concern in the early period. A stock of well-preserved virus was indispensable for the confirmation of reproducibility by other competing researchers. If infectivity was lost, it negatively affected the claim of “first” virus isolation.

Preservation of infectivity was a serious challenge in the early period of JE research, because at first little was known about the optimal conditions for a longer preservation. Although preservation of vaccinia virus used for vaccination at a low temperature was known before 1900 [36], the idea did not catch fire among microbiologists at large. At the turn of the 20th century, a notion that germs would be destroyed at a freezing temperature was still widespread. It is noted that only in 1928 was trans-oceanic shipment of YFV-infected tissues from Africa to North America made possible without a loss of infectivity by storing the specimens in a refrigerator of a passenger ship [37]. 

In the early part of the 1930s, lyophilization was found to improve virus preservation. However, invention of a practical freeze dryer for a prolonged preservation of virus was developed only in 1935 [38]. Furthermore, in the early 1930s, in most laboratories use of 30–50% glycerol at a low temperature (1–4 °C) was still the only available method of preservation for up to a few months. Preservation was important, because providing opportunities to other researchers for an independent confirmation of reproducibility of the results was an unwritten consensus. 

#### 3.2.8. Proof of Vector-Borne Transmission

This was an added requirement uniquely for vector-borne viruses, such as JEV. At the time, nearly all isolates were made of the specimens collected from patients, and causal relationship was established with these isolates. When JEV was isolated from mosquitoes in 1938 for the first time, the validity was seriously questioned by physicians because few of them could understand how a virus in mosquitoes could cause JE in humans. Thus, Tokushiro Mitamura and his colleagues [39] conducted laboratory experiments using infected mosquitoes and mice to prove vector-borne mode of transmission to vertebrates, to convince the skeptics. They also confirmed vertical transmission of JEV in mosquitoes in nature [40]. 

## 4. Brief History of the Early Occurrence of the Outbreaks of an Encephalitic Disease in Japan and Historical Accounts of the “First” Isolation of JEV

### 4.1. Brief History of Early Outbreaks

It has been speculated that relaxation of refrain from meat consumption among Buddhists in Japan at the end of feudal period in mid-19th century coincided with emergence of encephalitic outbreak. While it is generally agreeable, strictly speaking, swine breeding and popularity of the consumption of pork were recorded in Okinawa and southwestern parts of Kyushu for nearly a few centuries even during this period of refrain. In Okinawa, in particular, because of heavy influence of Chinese culture (including cuisine), consumption of pig meat and swine breeding had been a long tradition. Similarly in Kyushu, because of the influence of European visitors (including missionaries), pig breeding and meat consumption persisted even during the “refrain period,” albeit at a level below Okinawa’s. Because these are the regions of the country where earliest cases of JE typically begin to emerge at the beginning the season to start a dispersal to the northeast, any historical record of encephalitis outbreak there would suggest possible occurrence of JE much earlier. The fact that such a record was not found then favors the theory that JE first appeared only around 1871, although a possibility of lack of individuals interested in documenting diseases or loss of documents as an alternative explanation cannot be entirely ruled out. 

The earliest record of encephalitis outbreak with a considerable number of fatality was recorded in 1871 through 1873. An outbreak of equine encephalitis which occurred in 1897 is strongly believed to be a JE outbreak. The peak of epidemic occurred usually in August through September. Because of this seasonality, often the epidemic was called “summer encephalitis.” Subsequently, outbreak occurred in 1901, 1903, 1907, 1909, 1912, 1916, 1917, and 1919. Among those outbreaks, the 1912 outbreak was notable for high incidence in the elderly people and the case fatality rate (CFR) approximately around 60%. In the 1919 outbreak, more than 7000 cases were reported, again heavily among the elderly people. This is puzzling because JE is known to be primarily a pediatric disease today. An early age-stratified epidemiologic study of the encephalitic cases in the western region of the country considered endemic focus showed that the proportions in 1–10, 11–20, 51–60, and >60 yr groups were 8.5, 5.8, 21.1, and 40.6%, respectively [41]. The other statistics for a period (1927–1941) revealed that the proportions of cases in 0–6, 6–10, 11–20, 51–60, and >61 yr groups were 5.9, 9.4, 6.1, 15.0, and 42.6%, respectively [42]. In other studies, age distribution was bimodal (young and old). 

These reports contrasted to the predominantly pediatric pattern in Korea, where the seasonality and other aspects of JE were very similar to those in Japan. After WWII, even in Japan JE became predominantly a pediatric problem. It was suspected that the most likely reason for the unusually high ratio for the elderly people in Japan was because in public health reports, the statistics was then based on clinical diagnosis but not on laboratory confirmation [43]. However, the real reason may be more complicated. In the period (1970–2003) morbidity in many years dropped sharply in Japan to less than 100 after 1972 and less than 10 per annum after 1992. Besides the successful vaccination campaign, the other most likely cause of decline was a combination of change in agricultural practices and environmental changes [44]. In those lean years, the majority of the patients were again elderly. However, though sharing predominance of elderly patients, the statistics in two periods (1920s and 1980–2003) cannot be directly compared because of a significant difference in medico-ecologic background of the country. In Malaysia between 1940s and 1990s, the pattern of JE incidence was bimodal (children and old), while it is predominantly pediatric now [13]. In China, traditionally morbidity rate has been highest in children [45]. However, in certain areas, a proportional rise of incidence in old age group was recently observed. This was partly explained by the migration of youth to urban centers for jobs, which left disproportionally more elderly people living in the countryside where JE occurs more frequently. In other countries in Asia, predominance of pediatric pattern has been generally variable without a consistently fixed pattern, often depending on outbreak.

In the explosive 1924 outbreak, out of four main islands of Japan (Kyushu, Shikoku, Honshu, and Hokkaido in the direction from south to north) only Hokkaido was spared, as cases were recorded as north as Aomori of Honshu. After the 1924 outbreak, epidemic continued to occur. In the outbreaks of 1926, 1925, 1927, 1931, 1936, and 1937, CFR consistently exceeded 50%. Even in inter-epidemic periods, case number per annum often exceeded 600. It is interesting that in most of those years of outbreak in Japan, JE epidemic also occurred simultaneously in Korea with a very similar pattern of seasonality [43]. In much of northern half of China too, JE outbreak is also seasonal. 

### 4.2. Etiology of JE

Multiple theories were proposed on the etiology [41]. They ranged from Parkinsonism and herpes to encephalitis lethergica that prevailed between 1916 and 1927 [46]. “Virus of encephalitis of Japan” was proposed by Itsuma Takagi and “type B” was added to differentiate it from encephalitis lethergica of von Economo (type A) [41,47]. If the etiologies of the frequent and large-scale summer outbreaks in Japan and Korea between 1900 and 1930 were JEV, which is highly likely, the recent conclusion that the sharp increase in JEV population occurred in 1930–1960 [12] is incompatible with the historical data. This is another example of how insufficient amount of available literature negatively impacts on the accuracy of retrospective epidemiologic studies, which is commented in Section 6 and Conclusion later. 

### 4.3. “First” Isolation of JEV in Japan

As described earlier, the amount of historical documents made available to the scientists outside Japan (primarily to the scientists in the USA) in English represented only a small portion of the vast volume of publications in Japan. This is exemplified by the large numbers of articles covered in monographs published after the 1924 and 1935 JE outbreaks [41,48]. I provide a glimpse into this early JE research history, by focusing on “first” virus isolation conducted between 1920 and 1940. 

Between 1924 and 1938, three major outbreaks of JE occurred in Japan, though smaller numbers of clinical cases were continuously reported in inter-epidemic periods. In the current literature, a confusion is found, because of three different dates assigned to “first” isolation of JEV among authors. In Table 1, for each date (left column), the original references of isolation are listed on the central column, while the authors quoting the date are shown on the right column. 

**Table 1 viruses-14-02640-t001:** Three dates of “first” isolation of Japanese encephalitis virus.

1924	Ito et al. (1925) [49] Kaneko (1925) [50] Kaneko & Aoki (1928) [41] Kojima and Ono (1925) [51] Takagi (1925) [52]	Clarke & Casals (1965) [53] Erlander et al. (2009) [54] Le Flohic & Gonzales (2011) [55] Rivers (1927) [56] Tiroumourougane et al. (2001) [57]
1933/1934 *	Hayashi (1934) [58]	Endy & Nisalak (2002) [59] Halstead & Jacobson (2003) [60] ** Hayashi (2022) [61] ** Holbrook (2017) [62] Huang (1982) [63] Kobayashi (1959) [64] Okubo (1953) [23] Rosen (1986) [65]
1935/1936 *	Asami et al. (1935) [66] Hashimoto et al. (1936) [67] Kasahara et al. (1936a) [68] Kasahara et al. (1936b) [69] Kawamura et al. (1936) [70] Kobayashi et al. (1935) [71] Kudo et al. (1935) [72] Mitamura et al. (1935) [73] Mitamura et al. (1936) [74] Takagi (1935) [75] Takenouchi et al. (1935) [76] Taniguchi et al. (1936) [77]	Burke & Leake (1988) [78] European Centre for Disease Prevention & Control (Online Fact Sheets) (Year unspecified) [79] Innis (1995) [80] Karabatsos (1988) [81] Solomon et al. (2003) [82]

Bracket corresponds to reference citation number. * The date of actual “first” virus isolation/date of publication. Because some authors chose the date of publication (most of which 1 year later) rather than the date of actual isolation, the two consecutive years are treated to address the same event. ** The authors chose both 1933 and 1935 without a reference to which was “first”.

In the investigation of the 1924 outbreak, many workers representing multiple universities (i.e., Kyoto and Kyushu Universities, in particular) were involved in isolating the etiological agent [41]. Among multiple investigators who claimed to have isolated the agent and proved its filterability [41,49,50,51,52], Itsuma Takagi claimed isolation of as many as 6 strains after several passages in Asian monkeys and guinea pigs [52]. He further determined that the agent isolated could be preserved for 3 months in glycerine but lost infectivity when heated at 80 °C [52]. By the tradition of the country at that time, while the majority of the articles were published in Japanese, a considerable number were published in German [41]. However, because all strains eventually lost infectivity, no one could cross-examine to confirm the reproducibility of others’ claims. 

The tradition of quoting 1924 as the date of the “first” JEV isolation most likely originated in the publication of Thomas M. Rivers, then the world leading authority of newly emerged virology independent of the traditional bacteriology or microbiology. It should be noted that in his publication [56], he merely listed JEV as another filterable agent and that Rivers did not attempt to segregate physically isolated viruses from others (including JEV) which had not been. 

In the case of “first” isolation in 1933, Michitomo Hayashi inoculated multiple brain specimens from the deceased victims into the brain of Asian monkeys [58]. Among three brain specimens that induced a neurologic symptom in the first passage, only one could be passaged as many as several times. However, eventually infectivity of this isolate disappeared, and virus isolation was not completed. This report is basically similar to the aforementioned multiple inconclusive reports of the 1924 investigation but received more attention in the U.S. and elsewhere most likely because the report was published in both German and English. Still, as in the 1924 reports, confirmation of filterability was the only evidence. 

In the “first” isolation during the outbreak that affected multiple regions of the country in 1935, a large number of researchers representing governmental laboratories, universities, research institutions, and hospitals located in multiple locations participated in isolation. The word “virus” was officially adopted for the suspected agent during an emergency conference held that year. As shown in Table 1 [66,67,68,69,70,71,72,73,74,75,76,77], many strains were isolated. The investigators included a group from Kitasato Institute who reported isolation of “Nakayama” and other strains [68,69]. Mitamura’s group from University of Tokyo alone claimed isolation of as many as 22 strains [73]. “Kalinina” strain was isolated by a team from the St. Luke’s International Hospital [72]. The exact total number of isolates is unknown, due partly to an ambiguous style of reporting devoid of specifics (such as strain name or other types of identification) and partly to the difficulty of securing all original documents now. 

Because many strains were “first” isolated simultaneously during one outbreak, when the prototype of JEV was selected in the U.S. for a compilation of arboviruses and related zoonotic viruses [83], selection of Nakayama strain was not surprising, because the report of its isolation was well known in the U.S. for its publication in English and because of widespread sharing of the strain among multiple laboratories there (Figure 1). Thus, simultaneous isolation of multiple strains is added to the list of the causes of confusion over “first” virus isolation. Overall, though the exact reason why quoting authors differ in the choice of the “first” date of JEV isolation is not known, the variation partially suggests difference in the definition of “first” virus isolation among workers even in the modern periods. 

Only the records of shipment or transfer for the 1935–1950 period which were documented in publications are used. Other known transfers which were documented but deficient in data recording and other transactions without any record are excluded. Green: Shipment/transfer of Nakayama strain; Red: Shipment/transfer of all other strains. As a virus exchange, SLEV was shipped from L. T. Webster of the Rockefeller Institute of Medical Research (RIMR) to Univ. Tokyo and St. Luke’s International Hospital in 1936 and later to Niigata University. For many transactions across the Pacific both virus and antiserums from patients were shipped. The names in double quotation marks refer to strains. 

The International Catalogue of Arboviruses published by the American Society of Tropical Medicine and Hygiene [81] (now online as ArboCat at www.cdc.gov/arbocat/virusbrowser.aspx, accessed on 9 October 2022) has served for many years as an invaluable source of voucher specimens for arbovirologists around the world. Regarding the information on the 1935 strain submitted to the Catalogue at the time of virus deposition, I found it difficult to identify the strain partly because of the lack of specific strain designation in the references submitted at the time of virus deposition. 

### 4.4. Virus Exchange across the Pacific and Other Transfers of JEV Strains

The JEV isolation in 1935 elicited a special interest among the U.S. investigators of neurotrophic viruses, because St. Louis encephalitis virus (SLEV) had been isolated only two years earlier. To compare the properties of these encephalitic viruses, the investigators of two countries arranged virus exchange. As a result, Nakayama and other strains of JEV (as well as homologous antiserums from patients) were shipped to some laboratories in the U.S., while SLEV (and the antiserums) were shipped to laboratories in Japan. In each country, imported virus was further shared with other laboratories. Other shipments from Japan to the U.S. are known, but due to the troubling style of recording, only those transactions documented in publications between 1935 and 1950 [35,74,84,85,86,87,88,89,90,91,92,93] are illustrated in Figure 1. At the U.S. Army Medical Department Professional Service School (after 1953 renamed Walter Reed Army Institute of Research), in March, 1942, only 3 months after the Pearl Harbor Attack, Albert B. Sabin began to use Nakayama strain for initiating a JEV vaccine project. The task of completing the vaccine development became more urgent by the end of 1944 because invasion of the Ryukyu Archipelago (including Okinawa Island) was projected to take place shortly and protecting the soldiers to be deployed there was critical. Though soldiers and other military personnel were vaccinated and found protected and the efficacy evaluation of the vaccine was conducted in Japan after the war, Sabin was not entirely satisfied. As Sabin changed his focus to other viruses after the war, Joseph E. Smadel carried on the task of improving further the JEV vaccine in the Army, by using Nakayama strain adapted to growth in chick embryo [86,94,95]. However, the vaccine research was terminated in 1952 after an unfavorable review of the quality due to reduced immunogenicity; and the task of further improvement was handed over to a Japanese agency.

In the meanwhile, unbeknownst among most JE researchers on both sides of the Pacific, during the difficult time of devastation during WWII, all JEV isolates in Japan, including the original Nakayama strain, were lost there [61,64,90]. Accordingly, after the end of WWII, Nakayama strain in the U.S. was shipped back to Japan, as it was needed for vaccine manufacturing. Although the date is unknown, Kalinina strain also returned home after 1945. In addition to the transactions in Figure 1, Nakayama strain was shipped from the U.S. Army to the YF Research Institute in Entebbe, Uganda (M.C. Smithburn) in 1941 [96], while an unidentified strain was shipped from Kitasato Institute to Robert Koch Institute, Berlin (E. Haagen) in 1936 [97], when a concern over a suspected JE spread to Europe arose [98]. Although the exact records of transactions have not been found, by 1950 the American Type Culture Collection (a nonprofit independent organization) had JEV strains listed in the Viral and Rickettsial Registry. It was the source of Nakayama strain used as reference for the identification of JEV strains by the U.S. investigators of JE in Malaysia in 1951 [99,100], while the U.S. Army provided Nakayama strain to the University of Malaya in Singapore during the 1952 outbreak [101]. 

### 4.5. The Twisted Background of Nakayama-NIH Designation

Nakayama strain of JEV occupies a special place in the history of JE research not only as the “first” strain but also for serving as source for the only JE vaccine and material for research worldwide for nearly half a century. Obviously, for many researchers, the following history rarely told may be of interest. After returning home to Yobo Kenkyusho (translated as National Institute for Preventive Medicine but more popularly known by its sigla, “Yoken” in Japan), Nakayama strain was shared with another governmental public health institution dedicated to veterinary product quality inspection and belonging to the Ministry of Agriculture and Fisheries (known by another sigla, “Yakken”) and Kitasato Institute, the birth place. In the subsequent JEV research, researchers studying any difference among Nakayama strain derivatives in different laboratories designated the derivative held at Yoken as “Nakayama-NIH,” another held at Yakken as “Nakayama-Yakken,” and the stock held at the origin of the return trip (RIMR-NY) as “Nakayama-RFVI” [102].

It has been long thought that “NIH” affixed to Nakayama strain meant Yoken of Japan, because some leading JE researchers in charge of vaccine project in that institution preferred their professional affiliation be called “National Institute of Health” in publications, following the namesake designation in the U.S. However, it turns out that this “NIH” attached to the vaccine actually meant NIH of the U.S. because of the following background. During the post-war period, quality certification of JEV vaccine produced in the U.S. Army was the responsibility of the National Institute (still singular in 1947) of Health. The agency then issued procedural instructions and quality standards which must be met by the lyophilized vaccine produced by using chick embryo [103]. As U.S. Army discontinued further JEV vaccine research shortly and transferred it to Japanese government, Japanese workers still complied with the above US-NIH guideline. That was the origin of affixing “NIH” to Nakayama derivative. Ironically, chick embryo was soon replaced by mouse brain (for human usage) due to insufficient virus propagation in chick embryo. Thus, compliance to the original US-NIH quality standards was no longer necessary. However, the abbreviation (NIH) remained stuck to the Nakayama derivative when vaccine production in Japan began in 1954. However, chick embryo-adapted Nakayama strain is still used for the production of a vaccine for veterinary usage.

### 4.6. The Impacts of Increased Passage History and Change of Hands

When virus strain is passaged many times, increased mutation and possible phenotypic change are the major concerns. By the end of 1940s, the passage level of Nakayama strain exceeded well over 100. In fact, antigenic difference among the Nakayama strain derivatives was reported [64,104]. Further, when a strain is transferred multiple times to laboratories where other JEV strains are maintained for research, theoretical probability of inadvertent mislabeling or accidental cross-contamination increases, jeopardizing authenticity of the strain. This has happened, for example, in the prototype (Sofjin) of tick-borne encephalitis virus in Russia [105]. Accordingly, it is prudent to perform sequencing of any virus before use in experiment or vaccine lot production, as emphasized in an evaluation of the current Chinese JEV vaccine because the genotype III vaccine (SA14-14-2) is known for the genetic diversity [106].

### 4.7. Improvement in Virus Propagation and Storage

In the early period of research, virus transfer to another institution meant shipping infected tissue or even organs. Naturally, the quantity of infectious virus shipped varied greatly. Furthermore, shipping blocks of tissues was cumbersome and dangerous. To standardize the amount of infectious virus per shipment, to make storage of virus simpler, efficient and/or to make it much less dangerous, adaptation of virus to tissue or cell culture was explored. As a result of this effort, pooled supernatant of infected in vitro culture was dispensed in a fixed quantity to a large number of vials for an easy storage and shipment. JEV was one of the beneficiaries of this technologic improvement. Another benefit of this improvement was minimal need of passage [23,33]. 

## 5. Early Characterization of JEV

In this section, early attempts to characterize JEV and its mechanisms of transmission and persistence in nature are briefly summarized. These contributions are later compared with the applications of molecular techniques for respective subject in Section 7.

### 5.1. Classification of JEV

In the early period, the relationship of JEV among known animal viruses was initially determined on the basis of antigenic relationship. Additionally for JEV, the relationship with other neurotropic viruses (including rabies virus, poliovirus, lymphocytic choriomeningitis virus, and others) was investigated. Jordi Casals, initially using complement fixation technique and hyperimmune serums, clearly distinguished JEV from eastern equine encephalitis, western equine encephalitis, and louping ill virus but recognized a close relationship with SLEV [31]. He then applied in vivo neutralization test and expanded the list of viruses compared, by adding Russian tick-borne encephalitis virus and West Nile virus [35]. Later, a system of organized registration of viruses with emphases on antigenic relationship and voucher concept evolved; and JEV was assigned in Group B viruses [107], with Nakayama strain its prototype [83]. After further research on antigenic relationship, JEV was classified as a member in Group B antigenic Complex consisting of 10 viruses including Murray Valley encephalitis virus, SLEV, Usutu virus, and West Nile virus [108]. 

### 5.2. Question over Overwintering and Persistence of JEV in Temperate Regions

Traditionally, regarding JEV persistence in temperate regions, four mechanisms have been proposed: overwintering of JEV in vectors, overwintering in vertebrate hosts, repeated (if not annual) introduction of the virus carried by vectors blown by atmospheric current, and repeated (if not annual) introduction of the virus by infected birds. 

JEV in overwintering mosquitoes was recorded in Korea as well as in the semitropical Amami-Oshima and Okinawa Islands further to the south of Kyushu of Japan [109,110,111]. However, detection of the virus in “overwintering” mosquitoes in Korea and in southwestern parts of Japan was best characterized as a sporadic event because of inconsistency, depending on the year of study [110]. Furthermore, because the mean, coldest monthly temperature in January in Amami-Oshima or in Okinawa Islands surrounded by the warm ocean current is approximately 14 °C or 16 °C, respectively, “overwintering” there is considered a state of temporary quiescence over winter rather than true overwintering. In fact, some female mosquitoes were found to temporarily engage in blood-feeding when ambient temperature rose substantially. Obviously, the ecologic conditions for the quiescence period there are vastly different from freezing conditions in much colder northern regions, such as interior of Japan, interior and northern parts of China, or in Far East region of Russia. 

JEV was once isolated from pigs early in the season (presumably infected between April and June) in Hokkaido in 1986, which raised a possibility of viral overwintering in northern part of Japan [112]. Actually, JE outbreak in Hokkaido was not new at the time, because equine outbreak had occurred multiple times there, including those in 1947–1948 and 1966. Furthermore, a JE outbreak in goats was documented in Aomori, just south of Hokkaido across a strait, beginning in March of 1949. If the virus overwintered there, it likely did in infected mosquitoes, because other possibilities of virus introduction (by the arrival of infected migratory birds and overwintering of JEV in hosts) have never been proven, as explained elsewhere. Still, the possibility of sporadic overwintering of JEV in northern parts of Japan remains. 

JEV overwintering probably occurs sporadically in other regions as well, but the persistence of virus overwintering is most likely only for a short period of time. Additionally, in northern temperate regions, isolation of JEV in overwintering mosquitoes has been rare. This is not surprising because overwintering female mosquitoes generally do not blood-feed before hibernation. Accordingly, in the subsequent absence of any report of newly infected pigs in early spring in Hokkaido and elsewhere in northern temperate regions, it has been difficult to interpret the mode of transmission except to recognize occurrence of infrequent (but short) viral overwintering. On the other hand, repeated (if not annual) introduction of virus-infected mosquitoes blown by atmospheric air current has been more strongly supported by the sequence similarity of newly introduced genotype I JEV between the strains in the continental parts of Asia and those isolated in Japan, since genotype I had not existed there before [113].

As for vertical transmission of JEV in mosquitoes, it was discovered for the first time in 1938 [39,40]. It was subsequently found multiple times in China [63]. Venereal transmission in mosquitoes was also discovered in China [63]. Generally, these modes of viral transmission in mosquitoes were reported from southern half of temperate regions of the country. Thus, the conclusion that genotypes III and I-b of JEV are maintained in temperate parts (north of the Tropic of Cancer) of Asia by hibernating mosquitoes, vertical transmission, in cold-blood vertebrates and/or bats [114], if correct, may apply only to the southern temperate regions of China closer to the Tropic of Cancer and south of 30° N. Anyone who proposes overwintering of JEV in northern temperate regions as a mechanism of viral persistence must be able to explain why JE cases in humans or domesticated animals (principally pigs and horses) rarely occur in these regions earlier (well before July) in the season. 

Though almost totally overlooked for years, JEV transmission between larvae in aquatic conditions also occurs [115]. This mode of transmission should be investigated as another possible mechanism of viral persistence and amplification.

### 5.3. Roles of Hosts

Shortly after the first JEV isolation in 1935, an interest in other vertebrates as possible hosts of this virus grew. As a result, JEV was isolated from horses in 1937 [116], pigs in 1948 [117], cattle in 1948 [117], goat in 1949 [117], and from passerine birds in 1938 [118].

The importance of birds in JEV transmission was more clearly recognized during an extensive research by U.S. investigators (principally Edward L. Buescher and William F. Scherer) in Japan in the 1950s. They found (i) that ardeid birds are regularly and silently infected with JEV in the summer; and multiple strains of JEV were isolated from those birds. Because the viremia levels are sufficient for vector transmission, they concluded that bird-mosquito-bird cycle of transmission cycle in the summer was certain. They further speculated that viremic birds disseminated virus to mosquitoes in distant locations; (ii) that epidemic ceased with the arrival of cold season, resulting in concurrent cessation of mosquito biting activity; (iii) that they had no idea if and how JEV persists in temperate regions after summer and speculated the possibilities of overwintering in mosquitoes, latently infected hosts, or repeated introduction of virus by migratory birds at the beginning of each JE season, but without ever reaching a conclusion; and (iv) that, because of the complexity of involvement of multiple hosts, control of one amplifying host (such as pigs) would not be sufficient to suppress JE epidemic. However, they neither used the word “reservoir” nor speculated its identity. Furthermore, they did not state that mosquito-host-mosquito cycle was the only mechanism of JEV transmission [119,120]. Essentially, they proposed a working hypothesis that explained best the mode of natural JEV transmission based on available data at the time. However, they repeatedly cautioned against arriving at a definitive conclusion because of too many gaps in knowledge.

Although the role of pigs in JEV transmission has been a subject of strong interest recently, important first reports in history, such as virus isolation from pigs [117], discovery of high rate of stillbirth in pigs [121], confirmation of pigs as amplifiers [122], congenital infection in swine [123], identification of domestic pigs as a potential reservoir [124], and establishment of pig-mosquito transmission cycle under natural conditions as well as in laboratory experiment [125], have been rarely quoted in the modern literature. 

More recently, a report of nasopharyngeal contact transmission of JEV between pigs drew attention [126]. Though it was the first important report of its kind for JEV in pigs, this mode of direct transmission in pigs had been well known for many years for the transmission of other arboviruses, such as vesicular stomatitis virus (VSV) and African swine fever virus [127,128,129]. Continuous VSV transmission well into mid-winter in temperate regions of North America was observed when a chain of temporarily overlapping contact infection between infected and non-infected animals occurred. The shared environment for this direct transmission is crowded animal breeding facility, such as pen, barn, stable, piggery, duck house, or chicken coop. Because the recent report of nasopharyngeal contact transmission [126] was based on experiments, it is now necessary to determine if it is actually operating under natural conditions, since a mathematical modeling study concluded that the role of direct transmission between pigs could not be dismissed [130]. 

## 6. Communication and Information Retrieval

The aforementioned confusion over the “first” date of JEV isolation in the literature also derived from a language barrier in international scientific communication at the time. Because most early articles on JEV isolation were published in Japanese and because global bibliographic database on biomedical literature did not exist, few in other countries were aware of these publications. Accordingly, the publications in the early JE research found in nearly all bibliographic databases popularly used today are limited to a very small number of articles in English. Accordingly, the number of early JE papers retrievable from these databases is hardly commensurate with the actual number [131]. Furthermore, in meta analysis of the JE publications performed today, reference selection criterion chosen, such as Cockrane Review Criterion, automatically excludes non-English publications [132], further obliterating the early contributions written in languages other than English. Because the publications used for meta analysis are largely devoid of early JE contributions published in these languages, this popular method further facilitates a biased selection of scientific contributions. Thus, this is a sector of JE research where transition from early days to modern times badly failed.

Even in a more recent compilation of the past discoveries of arboviruses and zoonotic viruses around the world, which is otherwise considered most comprehensive thus far, all pertinent records including “first” isolations of JEV, DEN-1, Apoi virus, Yokose virus, Sagiyama virus, Akabane virus, Chuzan virus, and other viruses of veterinary importance from Japan and other arbovirus isolations in China and Korea are conspicuously and totally missing. For that matter, JEV research in whole Asia is not even mentioned [133].

The lack of interest in examining the documents written in languages other than English then led to excessive dependence on secondary or tertiary reading even when the originals were written in English. This increased the probability of the errors in quotation made by the earlier authors being perpetuated in JE literature. One example of frequent equivocal quotation relates to isolation of Nakayama strain from a 19 yr-old male. Actually, the strain was isolated from a 6 yr-old male, and the article was published in English [69]. 

Incorrect interpretation of published documents or reporting of incorrect information is another source of communication problem. One such example concerns the widespread belief in ardeid birds as reservoirs of JEV. As described in Section 5.3, a transmission cycle involving aedeid birds was proposed as a working hypothesis in the 1950s. What were needed in the following stage were studies to examine the scientific validity, either to prove or disprove. As described later, before all results were obtained for a scientific judgment, an unfortunate misinterpretation of those early reports somehow transformed the early hypothesis into a popular (but questionable) belief in the amplifier and reservoir roles of ardeid birds [134]. 

In another examples of communication problem, the source of the problems was often simple incorrect information. In one of such examples, it was reported that JE is “air or water borne, the results of mosquito bite or spread by ticks” [135]. While this was simply an unfortunate honest error, if scientific fact is deliberately distorted and used for a political reason, the consequence may be devastating. The tragic history in Soviet Union should not be forgotten. In 1937, two teams of researchers were dispatched to explore Far East part of the country to determine the cause of spring-summer encephalitis (TBE). Upon returning home after an arduous research in the field that yielded the first isolate of the virus, three members were arrested for an unfounded charge of disseminating JEV in the guise of performing a scientific research. Two of them spent 18 years in prison; and their professional careers were totally ruined [136].

## 7. Difference in Conclusion among Molecular Studies as a Rich Source of New Ideas and Research Questions as well as Sources of Cautions to Be Exercised

Development of metagenomics, phylogenetics, and phylogeography contributed to the enormous advancement in virologic research; and their advantages over traditional techniques in many aspects of investigation are numerous. Nevertheless, like in natural transmission studies of most vector-borne viruses still with multiple unresolved basic questions, differences in the results or conclusions among reports are also frequently found in these new molecular fields despite the fact that every author strives to present the best results and conclusion. Rather than frustrating over a lack of consensus, one can seize these differences in results or conclusions as excellent sources of innovative ideas and concepts to consider for further advancement. Thus, they serve as sources of research questions. They are also sources of cautions to be exercised. In this review, I emphasize accommodation of both traditional and modern techniques, since they complement each other. The three fields (metagenomics, phylogeography, and host range determination) discussed below partially overlap because they share genome sequencing as a necessary source of information. The topics selected below provide sources for comparison in JEV characterization between the early part of history and more recent age of molecular virology. 

### 7.1. Metagenomic (or Metatranscriptomic) Discovery of New Virus

In the past few decades, using a combination of metagenomics and bioinformatics, virus discovery has been revolutionized, resulting in the discoveries of a huge number of new viruses. One of the reasons for promoting metagenomic technique is that too many new viruses cannot be cultivated easily, which justifies use of culture-independent and sequence-independent detection/isolation method [18,137]. As demonstrated for Hepatitis C virus, even before the virus isolation, once sequence of a suspected agent was determined, therapeutic methods could be developed. Furthermore, unlike shotgun DNA sequence which generates an undesirable viral enrichment, such a bias does not occur with this new technique. Additionally, a huge diversity of RNAs in a given sample can be effectively surveyed. However, it is stressed that despite these spectacular successes and advantages, nothing has changed over years in the importance of the traditional requirement of physical isolation of the infectious agent as a requirement in “first” virus isolation. 

Despite the popularity of metagenomic virus isolation without a physical isolation of infectious entity, the application is considered problematic [138] because of multiple concerns. First, in the absence of infectious virus which can be shared with other groups, definitive confirmation by cross-examination required for a new virus is compromised. Second, the techniques for synthesizing new genomes have been available since the beginning of the 21st century [139,140], which may generate unintended consequences and immediately met a strong opposition for a variety of unintended possibilities [141]. Bioinformatics tools used are inadequate to reliably distinguishing metagenomically discovered viruses (which demonstrate little or no sequence similarity to the viruses in the database) from fake viruses artificially designed fraudulently [142]. For example, if someone of wicked mind artificially synthesizes a new virus, mixes its enzyme-digested genome fragments in a biospheric sample, and claims discovery of a new virus by metagenomics, it is difficult to determine if it was a honest or fraudulent report, in the absence of infectious virus that can be subjected to independent verification by experts unaffiliated with the author. Third, it is often difficult to tell if the novel viruses discovered by metagenomics are extant viruses or extinct viruses. Fourth, it is often difficult to build a phylogenetic tree, particularly when the new sequences are radically different from those of the existing viruses. It is emphasized that my concern applies only to those studies without an actual isolation of infectious agent, since in some metagenomics studies infectious agents were indeed isolated. To a certain degree, this new problem shares a similarity with some of the questions raised over the proof of “first” JEV isolation (Section 3). 

The fifth concern is that this new technique does not necessarily provide information regarding host range, tissue specificity, mode of transmission, virulence, and other biologic traits. Accordingly, even the proponents of metagenomics agree that the traditional and newer methods play complementary roles in virus isolation [18]. 

### 7.2. Phylogenetic Identification of New Viruses and Phylogeography of Flaviviruses

As in the aforementioned cases of metagenomically discovered viruses, new viruses are determined mostly by the position in phylogenetic tree supported by an assumed mathematical reliability of selected algorithm. Recently, International Committee on Taxonomy of Viruses (ICTV) endorsed a comparative genomics as a sole acceptable basis for the recognition of virus taxa [143]. Phylogenetic techniques have proven to be extremely useful and are popular among researchers. No one can deny the enormous benefits the molecular technologies have brought us. However, apart from the benefits, there are concerns. The concerns apply to “first” isolation of all viruses; but in this article the importance is more strongly directed towards the reports of insect-specific flaviviruses and the status of other questionable vector-borne flaviviruses. While statistical accuracy based on selected algorithm is often emphasized by all authors, the criteria used as a new species demarcation had not necessarily been accepted universally. Actually, in too many publications, nothing is mentioned about the criteria used. It has been known that multiple factors distort tree topology or reliability, such as too small number of or biased selection of sequences, missing links, selection of gene or genomic region, mechanisms of gene transfer, choice of outgroup (if used) and multiple other parameters [144]. Irrespective of the acceptance or rejection of concerns over these factors that may distort tree topology, it should be reiterated that phylogeny is essentially a branch of science of inference. As such, the goal of phylogenetic studies is the best approximation of the past history [145]. Because of the involvement of many assumptions for parameters and selection of software, naturally, there is no absolute guarantee that it depicts accurately the past history of the virus or the relationship among viruses. Furthermore, new or more advanced methods and concepts are continuously introduced, often generating different results. These are some of the causes of difference in results or conclusions [144].

A byproduct of phylogenetic studies is molecular clock. The complexity and problem of understanding phylogenetic data are shown in the following comparison of JEV molecular clock date determined with 95% posterior probability by three groups. It is stressed that comparing results of molecular clock dates among reports is not simple because of differences in software, assumptions used in parameter setting, and other analytical tools used. Nevertheless, it is generally agreed that genotype V is considered the oldest in JEV because it is located closest to the root of the phylogenetic tree. The dates determined for the genotype V were circa 1550s A.D. [146], circa 1902 (1813–1938) A.D. [114], and circa 1230 B.C. [12]. The huge difference in the third report was attributed to the inclusion of one more genotype V strain isolated recently in China, while only one strain was used in the first two reports.

For animals and plants, paleontological records (i.e., fossil, evolution of functional anatomy, geologic events, carbon dating), historical floral and faunal patterns of geographic distribution, and others can be used as collaborating supports of the accuracy of sequence-based phylogenies. For the taxonomy of bacteria, archaea, and eukaryotes, universally conserved genes serve as reliable guides for taxonomic classification. However, for nearly all phylogenies of naturally occurring RNA viruses, there are few or no such supports. For RNA viruses, RNA-dependent RNA polymerase gene (RdRp) is most conserved, which is why many RNA virus family trees are built on RdRp sequences. Still, the diversity of sequences in RdRp for many RNA virus groups is too great to be reliably used for the entire family. That is the fundamental weakness of RNA viral phylogenies. For vector-borne viruses, such as flaviviruses, it has been proposed to use endogenous viral elements (EVEs) integrated in the genomes of vectors, to offset the weakness. However, these vestigial signals are not found for all flaviviruses.

In the phylogeny of flaviviruses, the topologies based on NS3 and open reading frame (ORF) have been found to be very similar; while that of NS5 (including RdRp) was different [147,148]. For the NS3/ORF tree, thus far no collaborative evidence has been discovered. In an analysis of NS 5 tree, the topology was supported by two lines of collaborating evidence, sharing of motifs between two linked lineages along the branching order and the pattern of host or vector association change again along the branching order [149,150]. Furthermore, tree topologies based on prM-E, NS1, NS2A or NS2B, NS4A or NS4B genes, either alone or as concatenated structural genes or non-structural genes, generate aberrant topologies different from both NS3/ORF and NS5 trees [149]. Since ORF is concatenated with those genes, collaborating lines of evidence for the ORF tree need to be found, to enhance the reliability of this tree. They, in turn, will provide confidence in applying the current molecular criteria for the demarcation of virus species. 

In phylogeographic studies, accurate information regarding the date and location of each virus isolate is important. Furthermore, we need to recognize that inventory of available strains sequenced for a given study also changes the result considerably. Genotype V of JEV is a good example. Traditionally, the publication by a British physician, James H. Hale and two colleagues at the University of Malaya in Singapore (then a part of the Federation of Malaya) in 1952 [101] has been identified as the source of the only strain of genotype V (“Muar strain,” named for the location, Muar, Malaysia). However, in this 1952 article by Hale et al., the name Muar does not appear at all. Actually, Muar appears in Hale’s second Malaysian report published in 1954 [151]. In my survey of the JEV literature between 1990 and 2022, out of 6 most important phylogenetic articles in which Muar strain sequence was used because of its special importance in phylogeography or phylogenetics, none quoted the Hale’s 1954 article. 

Because the Muar strain had long been the sole strain of genotype V, it was once speculated that it represented “a transitional lineage between Murray Valley encephalitis virus and JEV, which subsequently became extinct” [82]. Unexpectedly, only 6 years after this speculation, one strain was isolated in China, followed by isolation of 5 more strains in Korea, necessitating change in our thinking. This illustrates how a conclusion in phylogenetics or phylogeography dynamically shifts over time, depending on the availability of samples at a particular time and new developments in research including introduction of new specimens, concepts and/or methods. 

Anyone accessing sequence data from GenBank notices that the person who isolated a strain earlier and the person who submitted its sequence many years later are different. Here lies a potential problem. As sequence data are customarily retrieved from GenBank or other databanks, associated information (such as date and location of isolation, etc.) deposited at the time of sequence submission is automatically used by most users of the GenBank on an assumption that it accurately corresponds to the isolate. However, when the submitters of sequences rely only on secondary reading, there is no guarantee of eliminating the possibility of misquotation. As an example of Muar strain of JEV, in one article Singapore was identified the geographic isolation site [152]. This is not an example of information retrieval problem mentioned earlier, since the Hale’s 1954 article in English is readable online free of charge. According to a phylogeographic report [55], data on JEV isolation have not been accurately documented in too many original publications. This seriously affects the interpretation and hence conclusions of phylogeographic studies. In some troubling articles, the direction of geographic spread of JEV follows the chronologic order of virus research. Thus, Japan is identified as the origin of virus spread to continental parts of Asia as if JE endemicity was well established in Japan. 

Another interesting subject that emerges from the issue surrounding Muar strain is the fact that Muar was the site of heavy fighting in 1942. Since JE outbreak began to become noticeable in Malaysia in 1942 [13], it is possible that JE dispersal began to occur in Southeast Asia as a result of war-related ecologic disturbance, as was the case of pan-Pacific spread of dengue in the aftermath of the battle in the Pacific during WW II. 

In another example of phylogeographic issue that generated contradictory results, based on E-gene sequences, it was earlier concluded that tick-borne flaviviruses dispersed westward from the east along the northern region of Eurasia [153]. When this topic was re-examined later but using full ORF sequences, it was concluded that these viruses spread both eastward and westward from a central point [154]. 

Theoretically, molecular data, if enhanced with biologic data, are more reliable. Actually, molecular data and phenotypic (biologic) data complement each other in the full characterization of a virus. Thus, if current molecular definition of viruses is considered two-dimensional, its combination with biologic traits makes it tridimensional. The following examples of flaviviruses are useful to understand it. Among multiple mosquito-borne flaviviruses (including JEV, Murray Valley encephalitis virus, West Nile virus (Kunjin virus), Zika virus and dengue virus serotypes) in tropical Asia, New Guinea and northern Australia, serologic cross reaction has been known. Existence of such non-specific antibodies in the human populations is not only a source of serologic complication in diagnostics but a factor that distorts efficacy evaluation of vaccines (for such diseases as JE and dengue). A recent report revealed a wide-spread interspecific proteomic recombination among those flaviviruses. Since this is not detectable in nucleotide or amino acid sequence-based phylogenies, it requires a different software [155]. This may partly explain the puzzle of cross reaction and represents an example of the benefit of a collaborative research between molecular virologic and biologic interests. Furthermore, this validates the importance of antigenic complex proposed earlier for flaviviruses [156]. 

### 7.3. Identification of the Hosts of Flaviviruses by Phylogenetics or Shared Molecular Determinants 

Identification of host (or host range) is an important subject not only for studying the evolutionary history of viruses but epidemiology caused by pathogenic viruses. From the days of poliovirus research in the 1950s, numerous discoveries of virus-specific receptors in hosts have been reported; and they established the foundation of the concept of RNA virus-specific host range. Most of these studies were performed in the laboratories, in particular for important human viruses. More recently, it has become popular to apply phylogenetics of viruses and bioinformatics data on host genomes to identify the hosts more precisely for known flaviviruses or to predict the hosts for newly discovered viruses based on sequence characteristics [157,158,159]. In the following discussion, metagenomically discovered viruses are excluded whether or not their hosts were identified. Four comments are presented.

First, when one attempts to correlate a viral genomic trait (or traits) with the corresponding host (or hosts), those viruses with the narrowest host range are most ideal. The examples include human-specific viruses, such as polio and measles viruses. For these viruses the virus–host relationship is direct with human the only host. On the other hand, for vector-borne arboviruses (such as JEV), it is indirect. Actually, on the parts of most vector-borne viruses, human is only one of the incidental hosts and unimportant for their survival. 

In host studies, many of the crucially important sources of information are found in voucher specimens collected in the field but not in the laboratories [160]. For most zoonotic viruses, shared denominator (or denominators) among human and many other animals including wildlife needs to be investigated. By focusing only on human, one may overlook other set of denominators shared among all hosts. Historically, voucher specimen concept evolved in the field investigations conducted by the Rockefeller Institute of Medical Research. As far as arboviruses are concerned, collection of voucher information for mostly prototype viruses was conducted until mid-1980s [81,138]; but regrettably the practice died thereafter in association with the rapid shift of interest among workers to molecular virologic research. According to a recent study, among viruses, only for yellow fever virus (YFV) a limited amount of collection of voucher specimens is still maintained in Brazil; but such a resource is very much limited for all animal virus groups [160]. Sequence files used in recent phylogenetic-bioinformatic host identification are based on sequences of the isolates mostly made in the past 3–4 decades. Thus, it is highly unlikely that they represent vouchered specimens. Accordingly, the reliability of the conclusions of these host studies is less certain. The reality that we lack a vital source of host information (voucher specimens) may surprise many readers, because we now live in the age of zoonoses and One Health in which shift in host association of viruses and spillover are repeatedly stressed as the major concern. Naturally, the lack of voucher specimens of SARS-CoV-2 is thought to be one of the major reasons why epidemiologic and virologic investigations have been hampered [160].

Second, recent host studies are all based on the traditional notion that vertebrates are the reservoirs. As revealed earlier, there exists no RNA arbovirus with a scientifically validated vertebrate reservoir. The only arbovirus with validated vertebrate reservoirs is African swine fever virus, the only DNA arbovirus. Thus far, the only validated reservoirs of RNA arboviruses are vectors themselves, based on the current definition of “reservoir” for arboviruses [161]. From the standpoint of a hematophagous vector-borne virus, because vectors are the reservoirs, vertebrate hosts are determined indirectly whether or not virus likes. Accordingly, the correlation between virus and its vector group depicted in phylogeny is more consistent and accurate than between virus and host group [162]. If a virus does not like a particular host, its survival depends on the alternative hosts its vector chooses. On the part of vectors, they must have a constant access to the sources of blood meal for survival. They can ill afford to obligatorily depend always on the hosts a virus likes best, because sufficient supply of blood meal from limited number of host species is not guaranteed in nature. Although the range of sources of blood meal is generally determined by innate feeding behavior of vectors, many vectors (such as vectors of JEV) also engage in opportunistic feeding, seasonally changing hosts depending on the availability and on other ecologic factors [163,164,165]. Thus, switching among birds, bovines, equines, pigs, wildlife, humans or even dogs is common among many JEV vectors. As an example, in JE-affected parts of India where pigs are only sporadically found, cattle were the source of nearly 90% of blood meals found in JEV vectors. If necessary, the vectors even feed on reptiles and amphibians. Thus, the direct relationship between arbovirus and hosts is weakened.

Third, many viruses are known to replicate in multiple phyla of organisms, some even in multiple kingdoms. A few Nodaviruses (Family *Nodaviridae*; not arboviruses), for example, replicate in mosquitoes, yeasts, vertebrates, and plants. Totiviruses replicate in fishes, mosquitoes, fungi, and protozoa. Mokola virus (not an arbovirus but a rhabdovirus affiliated to lyssavirus group) replicates in mosquito and vertebrate and has been often isolated from mosquitoes in nature. Similarly, members of the Genus *Flavivirus* are not only vector-borne and insect-specific viruses but viruses of nematodes, crustaceans, squids, sharks, and salmons. The vertebrate flaviviruses (such as Entebbe bat virus, Sokuluk virus, and Yokose virus) readily replicate in mosquito cells, even though they are not vector-borne; but, to make the matter further complicated, other group of vertebrate flaviviruses (known by No Known Vector viruses) do not replicate in arthropod cells. Accordingly, for those groups of viruses that have such a broad range of hosts, the accuracy of linking genetic signals found in sequence data of one host or one group of hosts to a particular virus or a closely-related group of viruses becomes less certain. 

It is emphasized that mere isolation of viruses from a particular organism does not establish a virus-organism relationship for a variety of reasons. A virus isolated could be a virus accidentally ingested by a host but remained infectious without replication [150]. Even if the virus isolated replicated in a host, identity of its true host remains unknown until the hosts the isolate had spent prior to isolation are determined. This is particularly important for the groups of viruses (including flaviviruses) that shift host frequently. 

The same precaution also applies to virus isolation from hematophagous vectors. Unnatural viruses of vertebrates or arthropods are sometime found in ticks or mosquitoes as a result of sucking blood of infected hosts, even though these viruses do not replicate in each group of respective vectors. The examples in ticks include rabies virus [166], corona virus [167], Tacaribe virus (an arenavirus) [168], murine herpes virus [169], and JEV [170]. Conversely, possible mechanical transmission of lymphocytic choriomeningitis virus [171], Burkitt’s lymphoma [172], Hepatitis B virus [173], and HIV [174] by mosquitoes was once topics of controversy. Generally, infectivity of exotic viruses is maintained far longer in ticks than in mosquitoes because of tick’s unique, very slow intracellular blood digestion mechanism called heterophagy and of the longer trans-stadial life span, compared with more rapid inactivation of exotic viruses in the hemocoel of mosquitoes. 

Fourth, innate host range potential of many flaviviruses is sometimes manifested in unnatural animals only under artificial conditions (such as, in captivity in zoo or aquarium). Some of the examples are infection of killer whale by SLEV [175], of alligators by West Nile virus [176], and of seals by JEV [177]. Additionally, several mosquito-borne alphaviruses readily replicate in the cell cultures derived from fishes [150]. These hosts, however, should not be considered natural hosts for the vector-borne flaviviruses.

Recently, a molecular marker, TMEM41B (a transmembrane protein), was identified as a pan-flavivirus host factor to explain difference in flavivirus infection among human populations in Asia [178]. However, a close examination of the report does not necessarily agree with the sweeping conclusion, given highly complex mixing of multiple ethnic groups in many countries. In other report, the role of birds is speculated to be involved in genotype shift of JEV [179]. The study is largely based on difference in replicative capacity in vitro among viral genotypes. As mentioned in Section 5.3, one problem is that exact role of migratory birds (i.e., ardeid birds) in JEV transmission has never been definitively proven, despite the initial excitement after the isolation of many strains from the birds [119], followed by an establishment of a wide-spread belief in avian role as reservoir. In reality, even the exact flyways of these long-distance migratory ardeid birds still remain unclear. Furthermore, mere isolation of virus from birds and detection in blood of anti-JEV antibodies alone do not satisfy avian role as amplifiers or reservoirs, though they confirm contact between the hosts and JEV. The findings of laboratory experiments need to be confirmed in the field. Despite the popularity of the avian role in transmission and the efforts to confirm it, subsequent field studies on ardeid birds in the tropics have been consistently negative [164,180,181].

Considering the complexities of host selection described above, it is necessary to understand host selection of vector-borne flaviviruses holistically in the context of the outcome of complicated interactions among viral, vectorial, host, and other ecologic factors. Actually, even the authors of one of those reports readily admit the difficulty of identifying hosts [157].

### 7.4. Viral Replication Assay and Persistence of Viral RNA without a Recovery of Infectious Virus

In recent decades, often viral assay has been performed by measuring viral RNA as a substitute for infectious unit assay. It is based on an assumption that copy number of viral RNA correlates with infectious unit (or viral load). This practice has been adopted in many branches of molecular studies including the analysis of the dynamics of viral replication. Although this correlation is valid when infectious unit assay is also performed as reference within the same set of experiments, when no such calibration of the quantity of viral RNA to infectious unit is conducted, the significance of the quantitative value of viral RNA alone is far less clear.

Prolonged RNA shedding has been suspected to be a possible evidence of direct transmission of JEV between hosts, JEV persistence or overwintering in temperate regions [182]. Actually, similar observations of viral RNA persistence have been reported for many arboviruses [161]. In a more recent report of direct transmission of JEV in pigs, viral RNA that persisted in tonsils exceptionally for as long as 25 days was infectious, while viremia period was less than several days [126]. Direct nasal transmission of JEV had been originally discovered in experiments using monkeys [183]. It was also known for other arboviruses such as dengue and yellow fever virus. One of the objectives at the time was to develop a vaccine strategy based on nasal application. In the past, two types of persistent JEV RNA were discovered, a longer defective interfering particle devoid of prM, E, and NS1 genes [184] and a much shorter RNA fragments in the 3′UTR [185,186]. The persistent RNA in the recent report [126] belongs to the shorter noninfectious 3′-terminal RNA on the basis of the genomic locations of the probes used. In all studies of persistent viral RNA, it is important to investigate what the detected viral RNA is, how it is generated, and what the significance of persistent viral RNA is. In a recent review of this subject, answers to those questions have not been obtained yet [187]. 

As for the significance of arboviral RNA persistence on overwintering, in all past reports of persistent detection of viral RNA in hosts through winter, when spring arrived, infectious agent has never been isolated from those hosts [161]. This includes failure to isolate infectious JEV in spring from pigs in which viral RNA persisted into winter [188]. Thus far, these results confirm that the persistent viral RNA generated or detectable in winter is not infectious. On the other hand, in human infections of central nervous system by neurotropic flaviviruses, infectious JEV and TBEV have been occasionally isolated from the brain of humans or nonhuman primates, after long periods. Though these cases demonstrate persistence of infectious virus, these examples cannot be used to support overwintering of the neurotropic viruses in humans or other mammals, since the viruses in the central nervous system, when released into the blood circulation, have been known to be quickly inactivated by neutralizing antibodies and did not participate in natural transmission by blood-sucking vectors.

In a more recent study, JEV RNA shedding into the vaginal mucosa of pig for as long as 28 days was reported; and the result was discussed in the context of possible sexual transmission of JEV in pigs [189]. Actually, sexual transmission of JEV in swine had been demonstrated much earlier in a simulated experiment, because viral shedding into semen had been observed during a JE epidemic [190]. The significance of these reports needs to be determined in two ways, theoretical possibility under laboratory conditions and practical importance in natural transmission. As for the latter importance, pig breeding by natural mating (pig mating) is practiced mostly in very small farms, while artificial insemination is more popular in larger breeding facilities where venereal transmission found under laboratory conditions does not apply. 

## 8. Concluding Remarks

This re-examination of the early history of “first” JEV isolation illustrates how the concepts and requirements for virus isolation and characterization evolved by trials and errors over time. While technological advancement has been phenomenal and the depth of our knowledge about viruses has increased considerably over years, some fundamental issues on natural JEV transmission that troubled the workers several decades ago still remain without a definitive conclusion. Additionally, this review revealed an enigma as to why such invaluable contributions to JE studies accomplished in the early part of history have remained untold for so long.

Among several troubling issues commented on, language barrier as a cause of international communication problem is unique to JE research. Though the exact background or reason is unknown, it is possible that before WWII, JE was primarily considered either a regional or domestic issue in Japan, even though JE was known in Korea, China, and Far Eastern Soviet Union. Thus, despite the fact that a large number of research reports were published in their native language, only a very small number of the workers published their reports in English. The traditional emphasis on German rather than English in science in Japan prior to WWII was possibly another reason. After WWII, however, JE became an international medical issue and English the global currency of scientific communication. Accordingly, had early historical JE research documents been systematically translated, ambiguities clarified and comprehensively published in English for the benefits to the researchers around the world, some of the communication problems, including the difference in “first” virus isolation date, could have been averted. In the future, it is highly desirable that each JE-endemic country (where English is not a mother tongue) post online a collection of selected, important research documents and publications which can be translated with a more advanced language software uniquely crafted for each language, to make the documents available worldwide.

As for the issues related to the strategies adopted to solve research questions, one common denominator among many proponents of molecular techniques is to analyze complex topics (ranging from virus taxonomy to host group determinants) by means of genomic and bioinformatics data alone. The advantage is that most parameters are quantifiable; and the support and popularity of this approach are considerable. In fact, a popular method for phylogenetic studies was once maximum parsimony, which was subsequently replaced by more improved methods. On the other hand, the proponents of holistic understanding of these complex issues regard such strategies too simplistic. Though they too prefer a simpler explanation if possible, for complex issues, they recognize complex explanations as rational and necessary choices. For the sake of healthy science, it is desirable that both sides of argument be kept alive. 

The history of international JEV-SLEV exchange (both virus and information) shown in this review may appear to some readers nothing but a fond memory now that, on top of the above problems, multiple reasons for restricting international (and even domestic) shipment of viruses have been erected as barriers due to a variety of necessities and concerns, such as increasing threat of bioterrorism, increased awareness of intellectual property right protection and even politics. Thus, cross examination of new viruses by other groups is far more difficult. 

When virus shipment is increasingly difficult, as it is today, at least sharing information can be improved. As an example, since 1960 through 1990th, American Committee on Arthropod-borne Viruses (ACAV) within the American Society of Tropical Medicine and Hygiene collected voluntarily-submitted research activity news and other developments from many laboratories around the world, printed them in an informal newsletter (“Arthropod-borne Virus Information Exchange”), and distributed it twice a year free of charge. Amazingly, many laboratories submitted preliminary results and other data still unpublished. A similar newsletter was also launched in Australia. What is sorely needed today is revival of such a collaborative and more open spirit in international scientific communication, this time online.

## Figures and Tables

**Figure 1 viruses-14-02640-f001:**
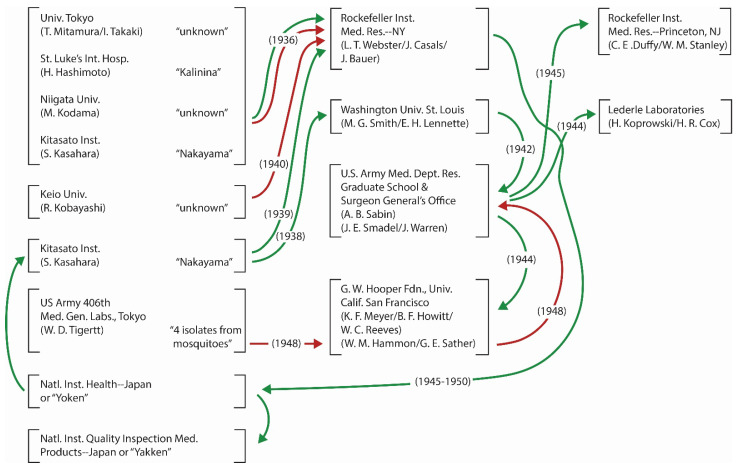
Diagram showing shipments and transfers of JEV strains between laboratories.

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
