# Peer review of "Contrasting the Practices of Virus Isolation and Characterization between the Early Period in History and Modern Times: The Case of Japanese Encephalitis Virus"

_viruses, 2022, doi:10.3390/v14122640_

Round 1

Reviewer 1 Report

Dear Author
Congratulations on your manuscript. This is a very interesting and comprehensive review on Japanese encephalitis virus. However, I believe a careful reading must be made in order to fix some small grammar typos. Also, I don’t know if the problem happened when conversion to pdf format was done, but, for example, attention should be paid to the scientific names of the mosquitoes which should be italicised. Also, the size of the letter is not uniform along the text.

Reviewer 2 Report

Kuno wrote a review paper about the isolation, appearance and spread of Japanese encephalitis virus in Japan, and the Far east. Other issues, data, problems questions concerning this topic are also discussed. It must be appreciated, that somebody look after decades-old epidemiological data about a zoonotic disease and write micobiological history. Such works could help in better understanding our present problems with Flaviviruses, and Flavivirus epidemics. Although, both the text and the title is too long and confused, not easy to follow. The whole review has not a backborne. For me it is not clear, what about the author really wants to speak. About the story of JEV epidemics at the Far-east? About the isolation of the first JEV strain, and its transport between different labs? At the end only discussing of phylogentic trees and metaenomics would worth a whole paper. So I was lost. The author should decide which is important, which is not. Where is point A where he starts from, and point B he is heading for. This paper must be re-edit, re-write, re-organize, because it is simply not followable as it stands. Recently my paper about the story of a Flavivirus epidemic in my country was accepted and it is just under publication. I can not write its details as a reviewer. I suggest to concentrate on the story and conclusion of the JEV epidemic in Asia. Evaluation and discussion of methods could stand as a separate publication. 

I have not checked the proper citing of references.

Suggestions:

Title: Should be shortened, and make it more clear and concise.

row 57- I would be very doubtful with such calculations. Several similar contradictory calculations were published about evolution of TBEV.

row 64- there is not necssary I think

rows 85-93. Usage of these terms should be summarized in one short sentence.

Chapter 3. rows-104-352

This is a very long not easy to follow part of the manuscript. It should be shorten and somehow a logical line of thoughts should be given. I felt being in a dark forest, reading this part. If the author intended to give some background information about the details, phrases of the next following parts, it should be given in a more clear, followable way. And much shorter, in soldier-like manner.

Chapter 4. rows. 307-329

This problem should be mention, but in short way. Recently I wrote a very similar paper (it will be published soon) about story of TBE epidemics in my country. I read early papers from different languages from the 1910s-1930s. I mentioned, touched the whole language, style, article types problem in 1-2 sentences.

Chapter 5.

rows 333-347. I can not see point in association of pork consumption to appearance of the virus in Japan. Why to spent sentences on unproved suppositions?

- rows 330-390.  Here I would summarize the geographical appearance and distribution of JEV in the far-east. In separate chapter disuss the age-ralated problem.

It worth mentioning that TBEV similarly to JEV 1924, appeared in the late 1920s early 1930s and caused more and more human cases both in Siberia and Western /central part of Europe.

-rows 354-355. Data indicate that TBE epidemic in Europe started as pediatric disease (Wallgren 1925), but it is not the case today.

-          rows 399-400. Fist publications were probably written in Japanese. How could they convinced Americans. Ourpapers from the same years unreadable and unacceptable for experts of a neighbouring country.

-rows 400-425. Contradiction(s) on the first proven isolation could be mention in 1-2 sentences only.

- rows 420-425. National pride is very important in believing in someone else’s result, if the publication was written in domestic not known language. The first appearance of TBEV in Europe is believed (mostly by German-speking scientists) in Austria (Schneider  1931). Although nobody read the original paper and contemporary Swedish, German, Hungarian publications are also available.

5.3. from row 472 +figure 1- I do not think, that so detailed description of Japanese-US laboratory virus transfer is necessary.

If the author thinks, that the story of the Nakayama strain is so important, the story could have been written more briefly in a separate chapter. Now parts of the story are hidden at various places of the paper.

rows 548-549. A recent article showed that the related Tick-borne encephalitis Flavivirus proved to be very stabile after hundreds of passages and laboratory usage for more than 60 years.

609-610.- In contrary to other issues, the author handles TBE virus in a very short, and not proper way. TBEV is the most important zoonotic Flavivirus in Eurasia. The virus was first isolated by a Russian-Soviet scientific expendition in Siberia during the 1930s. The first central European TBEV strain was isolated in Czechoslovakia, Beroun, the second in Hungary Tatabánya, 1954. The virus was divided into 3 subtypes (European Siberian, far-east) recently new subtypes were also established (e.g. Himalayan). This virus worth more than to spend two lines only with a 34 years old obsolete reference.

- row 636. chapter 7. Why the author turns suddenly from first isolations of flaviviruses to discuss metagenomics. The JEV issue is over?

-       row 638-650. Is this part necessary?

-       line 662. It is really very important to have the isolation in tissue culture. New isolations  should not be accepted only by genome data, PCr, or by metagenomics.

-       - 669-763. Tick-borne encephalitis was also appeared and spread in central Europe after a WWII events.

Spelling:

-          row 352 outbrakes

-          row- 476- the two contries

500- are excluded

row 637- letter size

Spelling-

Although in some places the data presented are too detailed, and partly unnecessary , it is not bad, that story, transport, isolation details of an important zoonotic pathogen is recorded somewhere.

Round 2

Reviewer 2 Report

It is still not easy to follow the paper, but it is much better than the previous version. TWo papers should have beeen written instead of this complex text.